# The Relation of Surgical Procedures and Diagnosis Groups to Unplanned Readmission in Spinal Neurosurgery: A Retrospective Single Center Study

**DOI:** 10.3390/ijerph19084795

**Published:** 2022-04-15

**Authors:** Caroline Sander, Henry Oppermann, Ulf Nestler, Katharina Sander, Michael Karl Fehrenbach, Tim Wende, Nikolaus von Dercks, Jürgen Meixensberger

**Affiliations:** 1Department of Neurosurgery, University Hospital Leipzig, 04103 Leipzig, Germany; henry.oppermann@medizin.uni-leipzig.de (H.O.); ulf.nestler@medizin.uni-leipzig.de (U.N.); michael.fehrenbach@medizin.uni-leipzig.de (M.K.F.); tim.wende@medizin.uni-leipzig.de (T.W.); juergen.meixensberger@medizin.uni-leipzig.de (J.M.); 2Institute of Human Genetics, University Hospital Leipzig, 04103 Leipzig, Germany; 3Independent Researcher, 12437 Berlin, Germany; katharina.sander@mailbox.org; 4Department for Medical Controlling, University Hospital Leipzig, 04103 Leipzig, Germany; nikolaus.vondercks@medizin.uni-leipzig.de

**Keywords:** spine surgery, unplanned readmission, index diagnosis, surgical procedure, surgical access

## Abstract

Background: Unplanned readmission has gained increasing interest as a quality marker for inpatient care, as it is associated with patient mortality and higher economic costs. Spinal neurosurgery is characterized by a lack of epidemiologic readmission data. The aim of this study was to identify causes and predictors for unplanned readmissions related to index diagnoses and surgical procedures. Methods: In this study, from 2015 to 2017, spinal neurosurgical procedures were recorded for surgical and non-surgical treated patients. The main reasons for an unplanned readmission within 30 days following discharge were identified. Multivariate logarithmic regression revealed predictors of unplanned readmission. Results: A total of 1172 patient records were examined, of which 4.27% disclosed unplanned readmissions. Among the surgical patients, the readmission rate was 4.06%, mainly attributable to surgical site infections, while it was 5.06% for the non-surgical patients, attributable to uncontrolled pain. A night-time surgery presented as the independent predictive factor. Conclusion: In the heterogeneous group of spinal neurosurgical patients, stratification into diagnostic groups is necessary for statistical analysis. Degenerative lumbar spinal stenosis and spinal abscesses are mainly affected by unplanned readmission. The surgical procedure dorsal root ganglion stimulation is an independent predictor of unplanned re-hospitalizations, as is the timing of surgery.

## 1. Introduction

To improve hospital care, the rate of unplanned readmissions within 30 days of discharge has emerged as a viable quality and performance marker [1,2]. An unplanned readmission represents an enormous resource burden in addition to high costs for the health care system. Therefore, identification of the factors that cause unplanned readmissions is essential [3,4,5,6,7]. Recent health care reforms, which penalized excessive readmissions financially, have brought the consideration of readmissions into the current focus. There is increasing interest in analyzing the predictive factors and consequences of unplanned readmissions [1,8].

However, a better understanding of unplanned readmissions is not only relevant from an economic point of view. A decrease in unplanned readmissions can also increase patient safety and satisfaction [9]. Unplanned readmission is associated with poor survival. In a recent study, Dickinson et al. showed that glioblastoma patients with unplanned readmission had a nine-month shorter survival [10].

Strategies to avert primarily preventable readmissions are essential. One measure to control unplanned readmissions is the transitional care program [11].

The predictors and causes for unplanned readmission are manifold, reflecting the heterogeneous patient population in neurosurgery [1,9]. Readmission rates in cranial neurosurgery have been found to be surgery- and diagnosis-dependent [12]. In the current literature, the majority of studies and predictor analyses were done in North America on patients with spinal disorders in neurosurgery [13]. However, transfer of conclusions or deduction of these recommendations to a German cohort with national differences in health care and in cultural composition of the population is not permissible [14,15]. Only few publications of German spinal neurosurgical groups on this topic exist, and up to now, no study has analysed the index diagnoses and the different surgical approaches in view of the 30-day readmission.

The aim of the present study was to assess prognostic factors in spinal neurosurgical patients undergoing surgical and non-surgical treatment in Germany. The secondary study objectives were to present the reasons for readmission and to stratify them into preventable and non-preventable reasons.

## 2. Materials and Methods

The internal review board of the Medical Faculty of the University Hospital Leipzig had agreed to this retrospective data analysis (167/18-ek). According to the approval of the ethics committee, the patient’s written consent is not required.

Administrative data from 1 January 2015 through 31 December 2017 of adult patients (>18 years) who had undergone neurosurgical treatment for spine disorders at the neurosurgical department were included in the monocentric, retrospective study. Unplanned inpatients at the University Hospital Leipzig within 30 days after the index treatment were identified. Patient readmissions were not followed when transferred to other hospitals. We excluded patients who were readmitted for scheduled reasons. The first set of “index admission” diagnoses contained all spinal neurosurgical disorders, and a subdivision into operative and conservative treatment was made. The patients were divided into the index diagnosis groups “degenerative”, “neoplasm”, “functional disorder”, and “other” (abscess, arteriovenous fistula, hemangioma). The classification was done according to the ICD-10 GM (see Appendix A Appendix A). The surgical procedures were presented according to the operation and procedure code list (OPS-list, see Appendix A Appendix A). For the observation period, we reviewed the hospital charts of each readmitted patient and obtained demographic information. Patient clinical complexity level (PCCL) was defined via the effective assessment ratio of the German diagnose-related group’s (DRG) coding level, which integrates the technical procedures and the patient’s secondary diagnoses. Three categories of readmission were defined: (1) preventable reasons (e.g., surgical site infections (SSIs), postoperative hemorrhage, nosocomial infection, postoperative pain, falls); (2) reasons despite best practice (e.g., recurrent herniation); and (3) unrelated reasons, as proposed in the literature [9].

Statistical analysis was performed with IBM SPSS Statistics 25.0 software (IBM, Armonk, NY, USA). For categorical variables, the Fisher exact test was applied; for more than two categories, the Kruskal–Wallis test was employed in the absence of normal distribution. Continuous variables were described using mean values, while categorical variables were described with counts and frequencies. Binary univariate and multivariate logistic regression tests were used to assess significant predictive factors. The threshold of metric variables was defined by the receiver operating characteristic. Factors associated with an unplanned readmission at the univariate level with a *p*-value of 0.20 or lower were integrated into the multivariate logistic regression model. A two-tailed *p*-value <0.05 was considered to be statistically significant.

## 3. Results

During the study period from 2015 to 2017, a total of 1172 patients were treated as inpatients in the Neurosurgical Department of the University Hospital Leipzig, of which 935 patients underwent surgery. The majority of patients belonged to the degenerative case group (879 cases, 75%). The exact demographic data of the patients can be seen in Table 1.

Of the 1172 patients included in this study, 50 underwent unplanned readmission for inpatient treatment. The characteristics of the readmitted patients are illustrated in Table 2.

### 3.1. Unplanned Readmission Cohort

The overall readmission rate of the study population is 4.27%. The majority of readmissions involved patients with degenerative spine disease (36 cases). The main reasons for unplanned readmission were SSI (17 cases), followed by pain (16 cases). The different reasons for readmission are shown in Figure 1.

The readmission rate among the surgical cases was 4.06%, with 38 unplanned readmissions. The main reason for unplanned readmission was SSI (15 cases), followed by recurrent disc herniation (three cases) and malfunction of implanted SCS electrodes (4 cases).

Among the above readmitted 38 patients with surgery at index admission, SSIs were present in 15 cases (39.5%), resulting in an overall SSI rate of 1.6% in surgical patients. Together with two SSIs in the conservatively treated patients, this sums up to a wound healing disorder rate of 1.5% (17 of 1172) in the whole cohort. SSIs in the non-surgical group are due to previous surgery before initial admission. The two SSIs in the non-surgical group refer to previous surgeries that occurred more than 30 days before re-hospitalization, and thus cannot be counted as index admissions in these cases.

SSI was superficial in 52.9% (9 cases), deep wound healing disorders were present in 41.2% (7 cases), and cerebrospinal fluid fistula occurred in 5.9% (1 case). One patient in the ‘surgical degenerative’ group underwent unplanned readmission at the Department of Vascular Surgery for an infrarenal abdominal aortic aneurysm, recorded in the ‘unrelated readmission’ category.

The readmission rate for patients treated conservatively or non-surgically was 5.06%, with twelve unplanned readmissions. Here, persistent pain (9 cases) followed by late-onset SSI (2 cases) was prominent.

A total of 31 cases required unplanned surgery; 13 SSIs, 3 SCS electrodes, and 4 recurrent disc herniations needed surgical revision. Among patients initially treated conservatively, six cases required surgery at the time of unplanned readmission. Four patients underwent surgery for persistent pain, one patient for SSI, and one for a herniated disc.

The majority of the 50 unplanned readmissions were classified as ‘preventable’ (35 cases, 70%), whereas 27 preventable cases could be assigned to the surgical group with SSIs (15 cases). For the non-surgical group, we identified eight ‘preventable’ readmissions, mainly due to uncontrolled pain.

Significantly more patients with lumbar spinal stenosis (42% vs. 26.1%, *p* = 0.021, Fisher exact test) underwent unplanned readmission. A detailed summary of the index diagnosis groups is shown in Table 3.

Considering readmitted surgical patients (*n* = 38), significant differences in the frequency of index diagnoses were detected compared with the non-operated group (*n* = 12). Patients with lumbar disc herniation were significantly less likely to be readmitted (15.8% vs. 31%, *p* = 0.048, Fisher exact test), whereas operated patients with lumbar spinal stenosis were readmitted more frequently (42.1% vs. 24.1%, *p* = 0.019, Fisher exact test).

The unplanned readmitted patients were treated for significantly longer during index admission than the group without readmission (index admission LOS 9–16 days: 29% vs. 13.4%, *p* = 0.014, Fisher exact test). The timing of index surgery was also different between the readmitted patients and patients that were not readmitted. Readmitted patients received significantly more frequent emergency night shift surgeries (5.3% vs. 0.1%, *p* = 0.005, Fisher exact test).

### 3.2. Operative Procedures

A closer look at the different surgical procedures and an analysis regarding the unplanned readmission group versus the non-readmitted cohort revealed only a few significant differences. The surgical procedures and access routes are shown in Table 4.

In particular, there were no differences between the two groups with regard to the surgical access. Regarding intraoperative procedures between the readmitted and the non-readmitted group, sequestrectomy (18.4% vs. 34.8%, *p* = 0.036, Fisher exact test) and intervertebral cage fusion (0% vs. 10.6%, *p* = 0.026, Fisher exact test) were performed less frequently and placement of an electrode for dorsal root ganglion stimulation was performed significantly more often (5.26 vs. 0.6%, *p* = 0.03, Fisher exact test).

### 3.3. Prognostic Factors

Binary logarithmic regression analysis was used to examine prognostic factors for unplanned readmission in a multivariate analysis. Considering the entire cohort, only surgery during night shift was shown to be predictive for unplanned readmission (Table 5).

Other factors, such as index surgical procedure, LOS, age, discharge modality, or pre-existing conditions, reached significance only at the univariate level (see Appendix A Appendix A).

A closer look at the surgical group allows the identification of further predictive factors. Lumbar spinal stenosis, spinal abscess as index diagnoses, and a dorsal root ganglion stimulation emerged as predictive factors. The presence of diabetes mellitus as a pre-existing condition was also an independent risk factor favoring unplanned readmission in this cohort.

Regarding the non-surgical group, PCCL and female gender were independent predictors of unplanned readmission. Further stratification for index diagnosis did not reveal any independent predictive factors. This section may be divided by subheadings. It should provide a concise and precise description of the experimental results, their interpretation, as well as the experimental conclusions that can be drawn.

## 4. Discussion

The present study examines not only the role of different index diagnosis groups, but also the influence of surgical procedures and access routes on the frequency of unplanned readmission within 30 days following index treatment. A pronounced heterogeneity of spinal pathologies with associated readmission rates and causes for readmission has been published previously.

The study discloses an overall readmission rate of 4.27% with discretely higher readmission rates in patients who were treated conservatively compared with operated patients. The literature reports similar readmission rates of 4% to 7%, depending on the index diagnosis in spinal neurosurgery [13]. Recently, in a German collective, lower readmission rates of 2% after spinal neurosurgical procedures were found [15]. However, this study had different patient numbers across the diverse diagnostic groups. In addition, we included non-surgically treated patients in order to represent a broader spectrum of patients in spinal neurosurgery.

Readmission rates vary by index diagnosis, with a readmission rate of 4.6% for lumbar [13], 2.5% for cervical degenerative pathologies [16], 7.4% for functional spine procedures [17], and 14.2% for spine tumors [18]. In our cohort, the readmission rate for spinal neoplasms was significantly lower. The readmission rate for spine patients after surgical treatment in the present study lies at 4.27%—much lower than the 7.4% rate in our corresponding local patient cohort receiving cranial neurosurgical treatment [12]. This difference can partly be explained by the higher complexity of cranial surgery and the longer cranial neurosurgical intervention times, as well as the higher need for inpatient treatment when complications arise [9]. Similar to cranial patients, SSI was found to be most important for readmission, whereas concomitant neoplasm or insufficient social support did not pose problems [19].

The readmitted operated spine patients showed a significantly longer index LOS than the group without unplanned readmission. This is consistent with previous studies that proposed to consider LOS as an independent predictive factor for re-hospitalization [20,21]. The need for repeated surgery is one source of prolonged LOS during index treatment; similarly, perioperative complications make an unplanned readmission more likely [22]. The main reasons for unplanned readmission was SSI (34% for the total collective and 39.5% for the surgical group), which is considered to be ‘preventable’. The overall SSI rate was low, at 1.4%, confirming previous publications. In other studies, SSIs were also found to be a leading cause of unplanned readmission, with a frequency of 24–39.8% [13,15,23,24,25,26].

Another potentially preventable reason for readmission was pain (32% of the whole readmitted group). Among surgical patients, this reason for admission was present in 18.42% and is comparable to reports in the literature [24]. Among the non-surgical group, a total of 75% of readmissions were due to uncontrollable pain. The reason for this appears to be inadequate control of symptoms by the initial conservative therapy. Especially in several patient groups with degenerative spine disease, surgical intervention is postponed until ineffectiveness of conservative therapy has been shown. This group of unplanned readmitted patients is thus difficult to minimize. An exact indication for neurosurgical intervention has to put the individual benefit of the patient into the forefront, bearing in mind differences in treatment regimes from a national, institutional, as well as interdisciplinary perspective [27].

Considering predictors, night-time surgery was found to be significant in the entire population. Night-time surgery increases the risk of intraoperative complications [28]. However, previous studies indicated that the timing of surgery did not affect the clinical outcome of patients [29]. Interestingly, in cranial neurosurgery patients, a relationship between the timing of surgery and the readmission rate was found, but it was restricted to patients with hydrocephalus [19]. However, in our study, the increased readmission rate for patients treated during night time could be due to limited available resources and, of course, the complexity and acuity of the case.

In addition, we demonstrated that the index diagnosis of lumbar spinal stenosis constitutes a significant predictor, mainly due to increased patient age and a higher number of secondary diseases. Especially, older patients after lumbar spine surgery have a higher complication and readmission rate [30]. Furthermore, spinal abscess as an index diagnosis was shown to be a prognostic factor for unplanned readmission. Most prominently, the surgical procedure of dorsal root ganglion stimulation was associated with a high readmission rate (7.4%). This is mainly due to infections or complications with the device [17]. Precise patient selection is thus mandatory to avoid postoperative complications.

In contrast to the present results, studies have shown that long-stretch lumbar fusion in particular is associated with increased readmission rates [18]. Dural tear and subsequent dural closure have been associated with an increased risk of developing postoperative complications [31]. Nevertheless, we could not substantiate a statistical association with increased unplanned readmission in this study. Rather, similar to cranial neurosurgical patients, patient-dependent factors, such as the presence of diabetes mellitus as a pre-existing condition, were associated with an increased risk of readmission [12]. Diabetes mellitus increases the risk of SSI after lumbar spine surgery [32] and results in prolonged LOS [33]. For the conservative therapy group, female gender was identified as a predictive factor for unplanned readmission. We consider this to be due to an increased PCCL in the female readmitted patients (PCCL at readmission of female vs. male patients: 2.87 vs. 0.87). The increased PCCL could be explained by a greater comorbidity rate, for example, diabetes mellitus and arterial hypertension, and different admission diagnoses favoring spinal neoplasms. In another study, gender was confirmed to be a risk factor for unplanned readmission, owing to a higher rate of diabetes mellitus [34].

In addition, we identified PCCL as a predictor, in accordance with previous studies [1,12]. PCCL represents a good measurement for the severity of the patient’s secondary diseases [35] and provides indirect information on the presence of complications during an inpatient stay, which can be taken into account preoperatively.

To minimize preventable unplanned readmission in spine surgery, we suggest an appropriate patient selection and strict indication. Similarly, postoperative wound care and patient education [36], as well as sufficient pain medication, can reduce unplanned re-hospitalization regarding the high impact of limited pain control and SSIs on preventable unplanned readmissions. Furthermore, based on our results, we do not recommend night-shift surgery, if the indication justifies waiting, because of reduced personal and resource capacities.

One of the limitations of this study is the retrospective single center design. Only patients who were readmitted to our hospital were identified. Patients with urgent emergencies such as thromboembolism or cardiac complications might have been admitted to hospitals elsewhere, near their home, and “small” wound problems may have been treated on an outpatient basis. Additionally, there is the limitation due to the coding function of medical documentation. It is not possible to compensate for the correctness of documentation or missing values.

Further studies with larger cohorts are necessary to estimate the reasons for readmission and risk factors of especially rare neurosurgical clinical pictures. In the heterogeneous field of neurosurgery, a targeted identification of risk constellations to avoid complications and unplanned readmissions is possible for some patient factors such as age, diabetes mellitus, or case complexity (PCCL) in connection with common diagnoses, e.g., lumbar stenosis, but remains elusive for the rare entities with individual operative approaches and risks.

## 5. Conclusions

We showed a total readmission rate of 4.5% in a three-year retrospective study of 1172 spinal neurosurgical patients in a large German neurosurgical clinic. Unplanned readmission is most often seen in patients who underwent surgery for lumbar stenosis or spinal abscess. Dorsal root ganglion stimulation was an independent predictive factor for unplanned readmission, as well as the side diagnosis of diabetes mellitus, the patient clinical complexity level, and night-time surgery. Furthermore, some predictors were partly modifiable, involving the surgeon, the anesthesiologist, and the patient’s generalist. The majority of readmissions were classified as preventable, with non-healing surgical wounds and uncontrolled pain being the most frequent.

## Figures and Tables

**Figure 1 ijerph-19-04795-f001:**
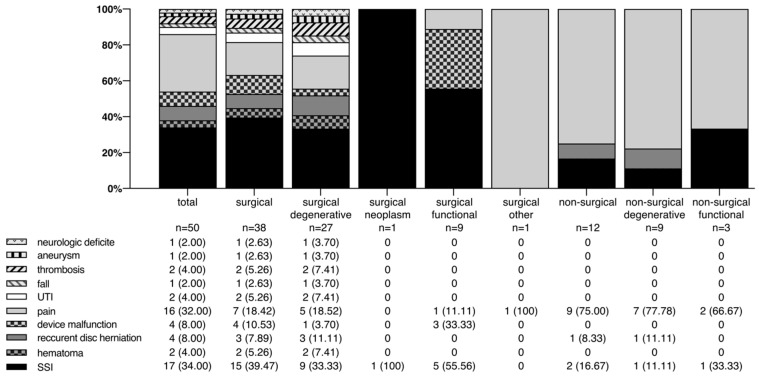
Reasons for unplanned readmission according to the index diagnosis groups. *n*, number; SSI, surgical site infection; UTI, urinary tract infection.

**Table 1 ijerph-19-04795-t001:** Characteristics of the total cohort.

Characteristics	*n* (%)
Index Diagnosis Group	
degenerative	879 (75)
cervical stenosis	82 (7.00)
thoracic stenosis	3 (0.26)
lumbar stenosis	312 (26.62)
cervical herniation	125 (10.67)
lumbar herniation	346 (29.52)
listhesis	11 (0.94)
neoplasm	62 (5.29)
unknown tumor	19 (1.62)
meningeoma	10 (0.85)
other benign tumor	17 (1.45)
malignant neoplasm	6 (0.51)
plasmocytoma	1 (0.09)
cyst	9 (0.77)
functional	192 (16.38)
chronic pain	186 (15.87)
spasticity	6 (0.51)
other	39 (3.33)
discitis	10 (0.85)
abscess	2 (0.17)
empyema	5 (0.43)
epidural hematoma	4 (0.34)
arteriovenous fistula	6 (0.51)
arteriovenous malformation	4 (0.34)
hemangioma	3 (0.26)
spina bifida	2 (0.17)
tethered cord	2 (0.17)
borreliosis	1 (0.09)
Patient Characteristics	
age, years	57.14 (18–92)
gender, female	540 (46.08)
PCCL, mean	1.54 (0.2–21.67)
LOS overall, mean in days	6.15 (1–68)
≤8 days, *n* (%)	988 (84.30)
9–16 days, *n* (%)	133 (11.35)
≥17 days, *n* (%)	51 (4.35)
number of side diagnoses, mean	4.37 (0–48)
high comorbidity ^a^	417 (35.58)
discharge	
home	1135 (96.84)
rehabilitation	22 (1.88)
external hospital	8 (0.68)
at own discretion	6 (0.51)
death	1 (0.09)
Surgical Characteristics	
surgery	935 (79.78)
surgery time, minutes	126 (19–561)
number of patients with ICU stay	32 (2.73)

ICU, intensive care unit; LOS, length of stay; *n*, number; PCCL, patient clinical complexity level. ^a^ High comorbidity: defined as five or more side diagnoses.

**Table 2 ijerph-19-04795-t002:** Characteristics of the readmitted cohort, stratified for index diagnosis groups and treatment management.

	Combined	Surgical	Surgical Degenerative	Surgical Neoplasm	Surgical Functional	Surgical Other	Non-Surgical	Non-Surgical Degenerative	Non-Surgical Functional
readmitted patients, *n*	50	38	27	1	9	1	12	9	3
readmission rate in %	4.27	4.06	3.82	2	5.96	3.57	5.06	5.23	7.32
total, *n*	1172	935	706	50	151	28	237	172	41
age, mean	61	63	65	59	57	59	57	56	56
gender, female	27 (54.00)	18 (47.37)	14 (51.85)	0	3 (33.33)	1 (100)	9 (75)	7 (77.78)	2 (66.67)
PCCL, mean	2.06	1.95	1.92	2.03	1.94	3.19	2.37	2.89	0.87
LOS, days	7.04	8.37	9.48	5.00	3.89	22.00	2.83	3.22	1.67
LOS readmission, days	7.90	8.92	10.44	4.00	3.67	20.00	4.67	5.11	3.33
time to readmission, days	13.30	12.11	11.52	30.00	10.33	26.00	17.08	16.22	19.67
surgery at readmission	31 (62.00)	25 (65.79)	17 (62.96)	1 (100)	7 (77.78)	0	6 (50)	5 (55.56)	1 (33.33)
reoperation rate in %	2.65	2.67	2.41	2.00	4.64	0.00	2.53	2.91	2.44

LOS, length of stay; *n*, number; PCCL, patient clinical complexity level.

**Table 3 ijerph-19-04795-t003:** Synopsis of index diagnosis groups comparing the readmitted and the non-readmitted population. Number, frequency, and *p*-value determined by Fisher exact test.

Total Cohort *n* = 1172	Non-Readmitted *n* = 1122	Readmitted *n* = 50
Index Diagnosis Groups	*n* (%)	*n* (%)
cervical stenosis	81 (7.22)	1 (2.00)
thoracic stenosis	3 (0.27)	0
lumbar stenosis	293 (26.11) *	21 (42.00) *
cervical herniation	121 (10.78)	4 (8.00)
lumbar herniation	337 (30.04)	9 (18.00)
listhesis	10 (0.89)	1 (2.00)
discitis	10 (0.89)	0
intraspinal cyst	9 (0.80)	0
unknown tumor	19 (1.69)	0
meningeoma	10 (0.89)	0
other benign tumor	16 (1.42)	1 (2.00)
malign tumor	6 (0.54)	0
hematoma	4 (0.36)	0
empyema	5 (0.45)	0
abscess	1 (0.09)	1 (2.00)
hemangioma	3 (0.27)	0
arteriovenous malformation	4 (0.36)	0
arteriovenous fistula	6 (0.53)	0
tethered cord	2 (0.18)	0
spina bifida	2 (0.18)	0
spasticity	6 (0.53)	0
chronic pain syndrom	172 (15.33)	12 (24.00)
other (borreliosis. plasmocytoma)	2 (0.18)	0

*n*, number. * *p*-value < 0.05.

**Table 4 ijerph-19-04795-t004:** Synopsis of surgical access and procedures comparing the readmitted and the non-readmitted population. Number, frequency, and *p*-value determined by Fisher exact test.

Combined *n* = 935	Non-Readmitted *n* = 897	Readmitted *n* = 38
	*n* (%)	*n* (%)
Surgical Access		
subcutaneous	15 (1.67)	2 (5.26)
intraspinal extradural	520 (57.97)	18 (47.37)
material implantation ^a^	312 (34.78)	18 (47.37)
intraspinal intradural	32 (3.57)	0
intraspinal intradural intramedullary	6 (0.67)	0
Surgical Procedure		
sequestrectomy and nucleotomy	312 (34.78) *	7 (18.42) *
spinal decompression	191 (21.29)	7 (18.42)
intervertebral cage fusion	95 (10.59) *	0 *
spondylodesis	71 (7.92)	6 (15.79)
spinal cord stimulation	59 (6.58)	4 (10.53)
generator implantation	41 (4.57)	4 (10.53)
resection of intraspinal tumor, except for neurinoma	31 (3.46)	0
resection of intraspinal neurinoma	14 (1.56)	2 (5.26)
wound debridement	13 (1.45)	2 (5.26)
medication pump implantation	12 (1.34)	0
peripheral nerve stimulation	10 (1.11)	1 (2.63)
resection of intraspinal empyema	10 (1.11)	1 (2.63)
resection of intraspinal hematoma	7 (0.78)	1 (2.63)
corporectomy	7 (0.78)	1 (2.63)
dorsal root ganglion stimulation	5 (0.56) *	2 (5.26) *
resection of arteriovenous malformation	5 (0.56)	0
lumbar puncture	3 (0.33)	0
biopsy	2 (0.22)	0
cancel/abort procedure	1 (0.11)	0

*n*, number. * *p*-value < 0.05. ^a^ Anterior cage fusion, spondylodesis, and generator and/or electrode implantation.

**Table 5 ijerph-19-04795-t005:** Predictors for unplanned readmission. Multivariate logistic regression for demographic data and hospital characteristics. Factors at the univariate level with a *p*-value ≤ 0.20 were integrated into the multivariate logistic regression model.

Multivariate Regression	OR (95% CI)	*p*-Value
Total Cohort		
age, >48 years	2.180 (0.674–7.057)	0.193
PCCL, >7	2.000 (0.870–4.595)	0.102
surgery, ≥2 interventions	1.776 (0.593–5.320)	0.305
night shift surgery ^a^	64.482 (4.270–973.702)	0.003
comorbidity ^b^	1.356 (0.646–2.848)	0.421
previous organ transplantation	12.054 (0.991–146.592)	0.051
index diagnosis groups		
lumbar herniation	1.672 (0.423–6.617)	0.464
cervical stenosis	1.842 (0.203–16.748)	0.588
lumbar stenosis	3.033 (0.988–9.310)	0.053
abscess	19.774 (0.980–398.907)	0.052
chronic pain	2.497 (0.761–8.197)	0.131
**Surgical Group**		
age, >50 years	1.926 (0.664–5.590)	0.228
LOS, >6 days	2.050 (0.880–4.775)	0.096
comorbidity ^b^	1.443 (0.656–3.174)	0.361
surgeries, ≥2 interventions	1.521 (0.473–4.896)	0.482
diabetes mellitus	5.284 (1.152–24.241)	0.032
previous organ transplantation	16.366 (0.973–275.302)	0.052
surgical access		
intraspinal extradural	1.047 (0.247–4.433)	0.951
surgical procedure		
sequestrectomy and nucleotomy	1.061 (0.250–4.502)	0.935
spondylodesis	1.954 (0.445–8.584)	0.375
resection of intraspinal neurinoma	4.768 (0.570–39.902)	0.150
generator implantation	3.826 (0.877–16.681)	0.074
dorsal root ganglion stimulation	11.665 (1.704–79.833)	0.012
index diagnosis group		
lumbar herniation	2.132 (0.367–12.402)	0.399
lumbar stenosis	3.874 (1.053–14.249)	0.042
abscess	28.482 (1.369–592.582)	0.031
benign tumor	6.241 (0.402–96.794)	0.190
chronic pain	3.054 (0.749–12.456)	0.120
**Non-Surgical Group**		
gender: female	5.915 (1.408–24.857)	0.015
PCCL, >1	4.404 (1.063–18.245)	0.041

CI, 95% confidence interval; LOS, length of stay; OR, odds ratio, PCCL, patient clinical complexity level. ^a^ night shift: 19:00 until before 07:00; ^b^ comorbidity: defined as five or more side diagnoses.

## Data Availability

The raw data of this work are available from the corresponding author (C.S.) upon reasonable request.

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
