# Peer review of "The Relation of Surgical Procedures and Diagnosis Groups to Unplanned Readmission in Spinal Neurosurgery: A Retrospective Single Center Study"

_ijerph, 2022, doi:10.3390/ijerph19084795_

Round 1
Reviewer 1 Report
The aim of this single center study from Germany was to assess prognostic factors in spinal neurosurgical patients undergoing surgery for unscheduled readmission. Secondary study objectives included reasons for re-admission and stratification into preventable and non-preventable causes. The study design is straight forward and the results are discussed in the context of the contemporary literature. The manuscript, however, requires substantial language/copy editing! Also, the introduction does not provide sufficient insights to the topic, and the aims provided in the intro need to be rephrased as the main objective is unclear.
Author Response
Reviewer #1
The aim of this single center study from Germany was to assess prognostic factors in spinal neurosurgical patients undergoing surgery for unscheduled readmission. Secondary study objectives included reasons for re-admission and stratification into preventable and non-preventable causes. The study design is straight forward and the results are discussed in the context of the contemporary literature. The manuscript, however, requires substantial language/copy editing! Also, the introduction does not provide sufficient insights to the topic, and the aims provided in the intro need to be rephrased as the main objective is unclear.
Dear Reviewer,
We thank you for your constructive criticism of our article. We have dealt with the points of criticism and hopefully implemented them to your satisfaction.
An extensive revision of the Introduction was made for better understanding and presentation of the topic.
Kind regard,
Caroline Sander
Reviewer 2 Report
I would like to thank you for submitting and give me the opportunity to review the manuscript entitled: “The relation of surgical procedures and diagnosis groups to un planned readmission in spinal neurosurgery. A retrospective single center study.” The research problem undertaken by the authors is interesting. I hope my comments will help to improve the quality of the manuscript in some way. Nevertheless, some questions and concerns need to be answered and corrected before the formal acceptance of the manuscript. In this sense, I only have a few minor comments.
In first place, it is necessary to add some studies and references in the introduction section. It is currently rather poor.
It is necessary to mention in the text whether the homogeneity and normality of the sample has been studied in order to justify the statistical tests applied. And for continuous variables, why don't you put the standard deviation?
The work has a very clear objective and that is “to assess prognostic factors in spinal neurosurgical patients undergoing surgery in Germany”. In this case, why, in the results, do the authors use the total cohort instead of using only the number of patients who underwent surgery? Similarly, table 1 and 3 go beyond the objectives of the study by looking at the total cohort. This fact leads to confusion in the interpretation of the results. Reconsider removing or reorganising table 1 by clarifying the results.
On the other hand, I consider that table 1 and table 2 should be located in material and methods section, instead of in results section because the data in these tables are not results but the sample itself.
On line 101 you write “Of the 935 patients receiving surgery, 50 had to be unplannedly readmitted for inpatient treatment.” In table 2 shows in the first column (combined), 50 patients readmitted out of a total of 1172. Explain the text or table 2 as the message is highly confusing. Idem at line 107.
Although, the authors could not substantiate a statistical association with increased unplanned readmission in this study, they have identified PCCL as a predictor of unplanned readmission. Many studies claim that secondary pathologies may be the cause of unexpected readmissions after surgery. In this case, why does this indicator not appear in the surgical group in table S3 of the supplementary material? Isn't this group the object of the study? What is the minimum value that this variable must have to be considered a predictor variable?
Review the bibliography. Some references with different formats.
Finally, some unimportant errors, such as in the footer of table 2 N is in upper case and in the table in lower case. Check it.
Author Response
Reviewer #2
Dear Reviewer,
We thank you for your constructive criticism of our article. We have dealt with the points of criticism and hopefully implemented them to your satisfaction.
I would like to thank you for submitting and give me the opportunity to review the manuscript entitled: “The relation of surgical procedures and diagnosis groups to un planned readmission in spinal neurosurgery. A retrospective single center study.” The research problem undertaken by the authors is interesting. I hope my comments will help to improve the quality of the manuscript in some way. Nevertheless, some questions and concerns need to be answered and corrected before the formal acceptance of the manuscript. In this sense, I only have a few minor comments.
In first place, it is necessary to add some studies and references in the introduction section. It is currently rather poor.
The introduction has been extensively revised.
It is necessary to mention in the text whether the homogeneity and normality of the sample has been studied in order to justify the statistical tests applied. And for continuous variables, why don't you put the standard deviation?
Statistical section was revised. Homogeneity testing was performed, unused tests were removed.
The work has a very clear objective and that is “to assess prognostic factors in spinal neurosurgical patients undergoing surgery in Germany”. In this case, why, in the results, do the authors use the total cohort instead of using only the number of patients who underwent surgery? Similarly, table 1 and 3 go beyond the objectives of the study by looking at the total cohort. This fact leads to confusion in the interpretation of the results. Reconsider removing or reorganising table 1 by clarifying the results.
The introduction was adapted. The special feature of this study should be the extensive group heterogeneity and the analsysis of several subgroups. Therefore, it is essential to include non-surgical patients (e.g. chronic pain in degenerative spinal diseases) in the study.
On the other hand, I consider that table 1 and table 2 should be located in material and methods section, instead of in results section because the data in these tables are not results but the sample itself.
In keeping with the structure of the paper, I advocate placing it in the Results section.
On line 101 you write “Of the 935 patients receiving surgery, 50 had to be unplannedly readmitted for inpatient treatment.” In table 2 shows in the first column (combined), 50 patients readmitted out of a total of 1172. Explain the text or table 2 as the message is highly confusing. Idem at line 107.
The information from the text was wrong, thank you very much for this very good hint. A change has been made.
Although, the authors could not substantiate a statistical association with increased unplanned readmission in this study, they have identified PCCL as a predictor of unplanned readmission. Many studies claim that secondary pathologies may be the cause of unexpected readmissions after surgery. In this case, why does this indicator not appear in the surgical group in table S3 of the supplementary material? Isn't this group the object of the study? What is the minimum value that this variable must have to be considered a predictor variable?
Numerous secondary diagnoses were included in the regression analysis. Those that had a p value <0.20 at the univariate level were included in the multivariate regression analysis. Therefore, for the combined group ‘previous organ transplantation’ and for the surgical group 'previous organ transplantation' and 'diabetes mellitus type II' were included for multivariate analysis. For the non-surgical group, there were no secondary diagnoses that had a p-value <0.2 at the univariate level. Overall, regression analyses were performed for the following secondary diagnoses for the above mentioned groups: ischemic heart disease, pulmonary heart disease, other heart diseases, cardiovascular disease, diabetes mellitus type II, high blood pressure, malignoma, kidney insufficiency, cerebrovascular disease, obstructive lung disease, HIV, previous organ transplantation, cardiac or vascular implants, obesity, malnutrition, anticoagulant therapy. If desired, the complete list of factors with results of univariate regression testing can be provided. This has been limited to only the statistically relevant factors due to the extensive factors.
Review the bibliography. Some references with different formats.
Formatting has been checked.
Finally, some unimportant errors, such as in the footer of table 2 N is in upper case and in the table in lower case. Check it.
The legend formatting has been revised.
Kind regards
Caroline Sander
Reviewer 3 Report
The authors want to identify causes and predictors for unplanned readmissions related to index diagnoses and surgical procedures in neurosurgical spine. The paper is well written, and the exposition is fluid and clear for the lecture. The goal of the research is for sure interesting and explore a underestimated field in surgical and neurosurgical branch, but central in health economy. Nevertheless, There are some minor critical issues that I'd like to address:
1) The introduction is valid, but lack of information about the impact of the unplanned readmission after surgery and its impact on health system. Please, stress more this point.
2) Method: Please specify the acronymous OPS and SSI
3) Results: The table 1 is quite confused. I suggest to split the table in 2, underling the diagnosis and the demographic characteristics.
In the introduction the authors affirm to have included patient undergone surgery treatment, but in the results they analyzed non surgical patients too. Why?
Many compliments for your outstanding work
Best regards
Author Response
Reviewer #3
Dear Reviewer,
We thank you for your constructive criticism of our article. We have dealt with the points of criticism and hopefully implemented them to your satisfaction.
The authors want to identify causes and predictors for unplanned readmissions related to index diagnoses and surgical procedures in neurosurgical spine. The paper is well written, and the exposition is fluid and clear for the lecture. The goal of the research is for sure interesting and explore a underestimated field in surgical and neurosurgical branch, but central in health economy. Nevertheless, There are some minor critical issues that I'd like to address:
- The introduction is valid, but lack of information about the impact of the unplanned readmission after surgery and its impact on health system. Please, stress more this point.
The introduction with regard to the relevance of unplanned readmissions for the health care system as well as patient safety was explained.
- Method: Please specify the acronymous OPS and SSI
OPS and SSI were explained in the methods section.
- Results: The table 1 is quite confused. I suggest to split the table in 2, underling the diagnosis and the demographic characteristics.
Table 1 has been given a clearer structure.
In the introduction the authors affirm to have included patient undergone surgery treatment, but in the results they analyzed non surgical patients too. Why?
The introduction was adapted. The special feature of this study should be the extensive group heterogeneity and the analsysis of several subgroups. Therefore, it is essential to include non-surgical patients (e.g. chronic pain in degenerative spinal diseases) in the study.
Many compliments for your outstanding work
Best regards
Kind regards
Caroline Sander